# Flame-Retarded and Heat-Resistant PP Compounds for Halogen-Free Low-Smoke Cable Protection Pipes (HFLS Conduits)

**DOI:** 10.3390/polym16091298

**Published:** 2024-05-06

**Authors:** Athanasios D. Porfyris, Afxentis Vafeiadis, Christina I. Gkountela, Christos Politidis, Georgios Messaritakis, Epameinondas Orfanoudakis, Silvia Pavlidou, Dimitrios M. Korres, Apostolos Kyritsis, Stamatina N. Vouyiouka

**Affiliations:** 1Laboratory of Polymer Technology, School of Chemical Engineering, National Technical University of Athens, Zographou Campus, 15780 Athens, Greece; vafeiadisafxentis@gmail.com (A.V.); cgkountela@mail.ntua.gr (C.I.G.); dmkorres@central.ntua.gr (D.M.K.); 2Dielectrics Group, School of Applied Mathematical and Physical Sciences, National Technical University of Athens, Zographou Campus, 15780 Athens, Greece; chrpolitidis@gmail.com (C.P.); akyrits@central.ntua.gr (A.K.); 3EMM. KOUVIDIS S.A. VIOPA Tylissos, 71500 Heraklion, Greece; georgios.messaritakis@kouvidis.gr (G.M.); panos.orfanoudakis@kouvidis.gr (E.O.); 4MIRTEC S.A., 76th km of Athens-Lamia National Road, 32009 Schimatari, Greece; s.pavlidou@mirtec.gr

**Keywords:** polypropylene, flame retardancy, halogen free, ageing

## Abstract

Conduits are plastic tubes extensively used to safeguard electrical cables, traditionally made from PVC. Recent safety guidelines seek alternatives due to PVC’s emission of thick smoke and toxic gases upon fire incidents. Polypropylene (PP) is emerging as a viable alternative but requires modification with suitable halogen-free additives to attain flame retardancy (FR) while maintaining high mechanical strength and weathering resistance, especially for outdoor applications. The objective of this study was to develop two FR systems for PP: one comprising a cyclic phosphonate ester and a monomeric N-alkoxy hindered amine adjuvant achieving V0, and another with hypophosphite and bromine moieties, along with a NOR-HAS adjuvant achieving V2. FR performance along with mechanical properties, physicochemical characterization, and dielectric behavior were evaluated prior to and after 2000 h of UV weathering or heat ageing. The developed FR systems set the basis for the production of industrial-scale masterbatches, from which further optimization to minimize FR content was performed via melt mixing with PP towards industrialization of a low-cost FR formulation. Accordingly, two types of corrugated conduits (ø20 mm) were manufactured. Their performance in terms of flame propagation, impact resistance, smoke density, and accelerated UV weathering stability classified them as Halogen Free Low Smoke (HFLS) conduits; meanwhile, they meet EU conduit standards without significantly impacting conduit properties or industrial processing efficiency.

## 1. Introduction

Cable protection pipes (conduits) are widely used in construction applications (residential, commercial, industrial settings, etc.) for enclosing, routing, and protecting electrical wiring, cables, or other types of utility lines and are provided as rigid or pliable (corrugated) conduits [1]. The current EU safety regulations for conduits [2] (are becoming stricter and demand a combination of properties for achieving flame retardancy [3], minimized smoke density [4], and as-low-as-possible halogen content [5], thus increasing the need for halogen-free and low-smoke (HFLS) compounds. Currently, the dominant material for the manufacture of conduits is PVC, which, though it exhibits fair flame-retardant behavior due to its inherent chlorine, nonetheless, upon burning, releases thick smoke containing toxic and hazardous gases, thus giving rise to safety issues for public health [6].

Therefore, PP emerges as a viable alternative, but as it is a highly flammable material, it requires halogen-free additivation for flame retardancy (FR) in order to comply with the aforementioned standards [7]. Furthermore, when it comes to surface electrical installations, additional UV and heat stabilization is needed, so as to increase its life cycle performance [8]. Last but not least, conduits exhibit demands in terms of mechanical properties, especially impact and compression, so as to efficiently facilitate protection to the contained cables. Consequently, it is anticipated that, for developing a PP compound designated for a bulky application like the manufacture of conduits, a puzzle of four key properties must be resolved, namely, flame retardancy, mechanical performance, halogen-free FR additives, and UV/heat stabilization [7,8]. In particular, the research challenge is to combine these two categories of additives without any antagonistic effect and at relatively low concentrations, so as to not significantly affect the processability and the mechanical performance of PP-based conduits [7].

Regarding readily/commercially available halogen-free flame retardants for PP, metal hydroxides and intumescent systems based on phosphorous are considered as good candidates [9,10,11,12,13]. On the one hand, metal hydroxides, e.g., Al(OH)_3_, are cheap and environmentally friendly FR solutions, operating mainly in the condensed phase by creating a thick inorganic char (e.g., Al_2_O_3_), which blocks further penetration of fire, but they also release water vapor (H_2_O) in the gas phase, thus diluting and cooling the flame zone [14,15,16,17]. On the other hand, intumescent systems comprising ammonium polyphosphate (APP), combined with appropriate char-forming agents such as pentaerythritol (PER) or triazine derivatives, also operate mainly in the condensed phase, and exhibit very attractive FR performance [18,19,20,21,22,23,24]. Nevertheless, when it comes to compounding with PP, loadings close to 50 wt.% are required for metal hydroxides and close to 20–30 wt.% for intumescent systems so as to reach V0 in the UL94V test. These FR contents are considered significantly high for retaining the mechanical behavior of the PP matrix. Moreover, APP is susceptible to hydrolysis during weathering [19,20], thus affecting the long-term performance of flame retardancy. To inhibit this, special protection is given to APP by microencapsulation or coatings [25,26], which, however, significantly increases the additive cost.

Another promising available FR additive category is the N-alkoxy piperidines (NOR-HAS), which can be incorporated at low loadings (1–2 wt.%) and offer FR performance usually aiming at V2 rating [9,10,16,27]. This additive family operates mainly in the gas phase by radical quenching and can act also as a long-term UV/heat stabilizer, since it generates nitroxyl radicals (NO·), well known for their stabilization effect. Due to the applied low concentration, the mechanical performance is not affected; nevertheless, when compounded alone, it is effective for film or fiber applications, but fails for bulk applications, such as conduits. However, it results in remarkable increase of FR performance and weathering resistance when used along with other FRs, such as intumescent systems [28,29].

Therefore, focus was given to readily available FR additives that can provide FR properties in as-low-as-possible concentrations, so that the PP mechanical properties are retained. The performance of these FRs was enhanced by the addition or presence of NOR-HAS additives, aiming at a simultaneous weathering resistance. Accordingly, two different FR formulations were examined, at a loading level of 11 and 4 wt.% respectively. The first system involves a cyclic phosphonate ester and a NOR-HAS derivative in a weight ratio of 10:1 [30]. Cyclic phosphonate esters are reported to operate simultaneously in the gas phase by releasing phosphorous-containing radicals, which in turn quench other highly reactive radicals (O**^·^**, OH**^·^**, CH_3_**^·^**) that promote fire, and in the condensed phase by char formation [31]. When applied alone, they require loadings of ca. 15 wt.%, reaching V2 classification, but if used along with NOR-HAS or sulfenamide compounds, the loading level is significantly decreased to ca. 10 wt.%, but most importantly, V0 rating is reached, since the observed dripping does not lead to cotton ignition [30,31,32].

On the other hand, the second FR formulation examined contains a specially designed readily available additive at the lowest level of 4 wt.%. The particular additive comprises mainly of aluminum hypophosphite (AHP), a NOR-HAS adjuvant that commences dripping offering at the same time UV/heat stabilization, and a phosphorous-bromine salt. The system operates, again, in both the condensed and the gas phase; the AHP decomposes to a pyrophosphate char, while releasing gases such as PH_3_ and H_2_O [33,34,35,36]. Moreover, from the decomposition of the phosphorous-bromine salt, phosphorous (PO**^·^**) and bromine radicals (Br**^·^**) are released in the gas phase and quench other flame-promoting radicals [37,38]. The latter effect seems contradictory, since only halogen-free alternatives are examined; nevertheless, the current standard for halogen-free conduits [5] permits a bromine content of up to 1500 ppm. Therefore, the second FR system lies far below the permitted halogen limit, and thus can be also considered as halogen-free [5], while the first FR system developed is actually a zero-halogen compound, since no halogen functionality is included.

The developed FR compounds were initially characterized in terms of flammability (UL94V test), mechanical properties (tensile and impact), and physico-chemical properties (MFR, DSC, TGA) so as to evaluate their performance as potential industrial HFLS conduit compounds. Moreover, BDS spectroscopy was applied in order to determine any potential changes in the semicrystalline morphology and/or molecular mobility of the amorphous part, that could be triggered by introducing the specific FR additives. Subsequently, they were subjected to separate heat and UV/humidity ageing tests for up to 2000 h, and their weathering resistance was evaluated in terms of flammability and mechanical properties. The lab-scale FR formulations were upscaled to industrial level by the development of the respective masterbatches (MBs). Finally, the production of new optimized FR compounds of reduced FR loadings and similar performance was attempted by melt mixing of the received MBs with appropriate amounts of reference PP. Accordingly, for the first FR system, V0 rating was feasible at a total FR loading of only 8 wt.%, while for the second FR system, V2 classification was reached at 3.2 wt.%, but most importantly, the bromine content was only 784 ppm, far below the halogen limit of EN50642 standard [5], i.e., 1500 ppm. The overall performance of the developed compounds rendered them applicable for conduit production trials. In fact, from real industrial scale tests, two (2) different types of corrugated conduits of ø20 mm outer diameter were developed and tested prior to and after an accelerated weathering ageing of 2000 h in terms of flame retardancy and impact properties.

## 2. Materials and Methods

### 2.1. Raw Materials and Additives

An extrusion grade heterophasic PP copolymer (ISPLEN, PB131N5E, Repsol, Madrid, Spain, MFR = 1.3 g/10 min) of high-impact strength was selected as the polymer matrix. Typically, the iPP block-copolymers, i.e., heterophasic copolymers, comprise ethylene-propylene monomer (EPM) or ethylene-propylene-diene monomer (EPDM) [39]. In addition, a homopolymer PP grade (ECOLEN HZ40S, Hellenic Petroleum, Athens, Greece, MFR = 25 g/10 min) was used as carrier for developing the industrial scale masterbatches. The readily/commercially available FRs tested here were a cyclic phosphonate ester (ADD1, Aflammit PCO900, THOR Specialities UK Limited, Northwich, England), a monomeric N-alkoxy hindered amine grade (ADD2, Flamestab NOR116, NOR-HAS, BASF SE Ludwigshafen, Germany), an FR system comprising aluminum phosphinate, a NOR-HAS adjuvant and a phosphorous-bromine salt (ADD3, Phoslite B713A, Italmatch Chemicals, Genova, Italy), and an FR mixture of phosphinates, e.g., aluminum and/or calcium (ADD4, Phoslite B85AX, Italmatch Chemicals, Genova, Italy). The ADD3 compound contains 2.8 wt.% bromine, while ADD4 is completely halogen free, according to the manufacturer. Chemical structures of the additives used are given in Appendix A.

### 2.2. Compounding

Two flame-retarded formulations will be thoroughly discussed (FR1, FR2, Table 1). FR1 comprises the cyclic phosphonate ester (ADD1) and the monomeric N-alkoxy hindered amine grade (ADD2) in a 10:1 ratio and is completely halogen free (zero halogen). On the other hand, FR2 contains ADD3 and ADD4. PP (ISPLEN, PB131N5E) and the desired additives of each formulation for the lab-scale compounds were melt mixed in a twin-screw extruder (Haake PTW16, Thermo-Fischer, Waltham, MA, USA, L/D = 25) in a temperature profile of 190-200-200-210-210-220 °C from hopper to die and screw rotation of 50 rpm. The melt temperature was ca. 225 °C for FR1 and ca. 224 °C for FR2. The average torque recorded during compounding was ca. 80 Nm for FR1 and ca. 66 Nm for FR2. The extruded material was cooled in a water bath and then pelletized. The received compounds, prior to any characterization or further molding, were dried in vacuo at 80 °C for 4 h. The compounding of the industrial scale masterbatches (MBs) was performed by the premix compounding process in an industrial twin-screw extruder by Plastika Kritis S.A (Heraklion, Crete, Greece), using PP carrier as the base polymer (with much lower melt viscosity), so as to overcome potential torque issues that would be triggered by the very high additive content. Finally, the development of the lab-scale FRMB1-FRMB10 compounds derived from the industrial scale MBs (Table 2) was performed according to the aforementioned extrusion conditions of FR1 and FR2. The difference in these compounds is that PP (ISPLEN, PB131N5E) is melt mixed with the respective amount of each industrial MB and not directly with the additives, thus better mimicking the industrial corrugation processing of the manufacturer.

### 2.3. Artificial Ageing Tests

The weathering resistance of the developed compounds was determined by separate artificial UV and heat-ageing tests. Regarding accelerated artificial UV ageing, samples were placed in a climate test cabinet (Nüve TK120, Nüve, Ankara, Turkey) following the conditions according to ISO 4892-3 standard [40] with UVA-340 (type 1A) fluorescent type lamps and were exposed to repetitive cycles of 8 h irradiation at 60 °C and 4 h condensation in the dark at 50 °C and humidity of 85% RH (Method A, Cycle No. 1). The total exposure of the samples was 2000 h, with 4 sampling intervals every 500 h. In each sampling interval, 5 impact specimens and 5 UL94 bars were removed from the chamber and characterized. On the other hand, regarding the accelerated heat ageing tests, tensile and UL94 specimens were placed in air circulated oven (Memmert ULE600, Memmert GmbH, Schwabach, Germany) operating constantly at 110 °C for 2000 h. Again, there were 4 sampling intervals every 500 h, during which specimens were removed from the oven. Finally, the accelerated ageing of the corrugated conduits was performed, according to the ISO 4892-2 standard [41], in a UV chamber equipped with Xenon arc lamps for 2000 h. Cycles of 102 min. dry and 18 min. water spray were performed, while the black panel temperature was controlled at 63 °C (Method A, Cycle No. 1, exposure cycles with temperature control by black-standard thermometer).

### 2.4. Characterization Methods

#### 2.4.1. Fourier Transform Infrared Spectroscopy (FT-IR)

FTIR was performed on a Bruker Alpha II spectrometer (Bruker Corporation, Billerica, MA, USA) using the ATR method with a diamond crystal in the range of 400 to 4000 cm^−1^ wavenumber region and a resolution 4 cm^−1^. Spectra were received for initial and aged FR compounds in the form of pellets, while each sample was analyzed in triplicate and the average spectrum was procured.

#### 2.4.2. Flame Retardancy Tests

The flame retardance performance of the produced FR compounds was determined according to UL94V. For each compound, 10 specimens of 125 × 13 × 1.6 mm^3^ were prepared by compression molding at 200 °C and ca. 200 bar and measured according to the standard. The UL94 bars were weighed prior to and after the fire test, so as to determine the mass loss of the burnt byproducts, i.e., produced gases and dripping of the material. In addition, regarding the developed corrugated conduits, flame retardancy was evaluated according to EN IEC 61386-22:2021 standard (resistance to flame propagation) [2]. Moreover, the smoke density behavior of the produced corrugated conduits was evaluated according to EN IEC 61034-2 standard [4]. The particular standard is designated for cables, but the methods included therein are used also for conduits as the most relative ones.

#### 2.4.3. Thermal Properties

All compounds were characterized in terms of DSC and TGA analysis. Regarding the DSC measurements, heating-cooling-heating cycles from 30 to 210 °C at a heating (cooling) rate of 10 °C/min were performed in a Mettler DSC 1 STARe (Mettler-Toledo International Inc., Greifensee, Switzerland) system under nitrogen flow (20 mL/min). From the cooling curves, the melt crystallization temperature (*T*_c_) and the corresponding enthalpy (Δ*H*_c_) were determined. The mass fraction crystallinity (*X*_c_, %) was computed-based on the cooling cycle (Equation (1), Δ*H*_0_ = 209 J/g [39], *ϕ* the additive nominal mass fraction in the compound). From the 2nd heating curves, the melting point (*T*_m_^2^) was determined.
(1)$$X_{c}=100\times \frac{{\Delta H}_{c}}{{\Delta H}_{0}\;(1-\varphi )}$$

In addition, the oxidation onset temperature (OOT) of the compounds was determined by separate DSC measurements from 30 to 300 °C at a heating rate of 10 °C/min under air atmosphere of 50 mL/min. Regarding the TGA analysis, all compounds were heated from 30 to 600 °C at a heating rate of 10 °C/min in a Mettler Toledo TGA/DSC 1 HT instrument (Mettler-Toledo International Inc., Greifensee, Switzerland) under nitrogen flow of 10 mL/min, and from the received mass loss curves, the temperature at 5% of weight loss (*T*_5%_), the temperature at the maximum weight loss rate (*T*_d_), and the final residue at 600 °C (R, %) were determined. All thermal properties measurements were performed in triplicates.

#### 2.4.4. Dielectric Properties

Broadband dielectric spectroscopy (BDS) measurements were performed in the temperature range 253–333 K, at atmospheric pressure, and for frequencies in the range from 10^−1^–10^6^ Hz. Measurements were made with a Novocontrol Alpha frequency analyzer (NOVOCONTROL Technologies GmbH & Co. KG, Montabaur, Germay) composed of a broadband dielectric converter and an active sample hand. The DS measurements were carried out in the parallel plate geometry capacitor and all samples were measured in the form of thick films (t~1–2 mm). In all cases, the complex dielectric permittivity *ε** = *ε*′ − *iε*″, where *ε*′ is the real and *ε*″ is the imaginary part, was obtained as a function of frequency *f* and temperature *T*, i.e., *ε**(*T*, *f*) [36]. The analysis of the DS curves was made using the empirical equation of Havriliak and Negami (HN, Equation (2)) [42]:(2)$$\varepsilon_{HN}^{*}\left(f,\;Τ\right)=\varepsilon_{\infty }\left(Τ\right)+\sum_{k=1}^{3}\frac{{\Delta \varepsilon }_{k}(T)}{{\left[1+{\left(i2\pi f\tau_{HN,k}(T)\right)}^{m_{k}}\right]}^{n_{k}}}+\frac{\sigma_{0}(Τ)}{i\varepsilon_{f}2\pi f}$$

Here, *k* indicates the process under investigation, Δ*ε*_k_(*T*) is the relaxation strength, and *τ*_HN,k_ is the relaxation time of the equation; *m*_k_, *n*_k_ (0 < *m*_k_, *m*_k_*n*_k_ ≤ 1) describe the symmetrical and asymmetrical broadening of the distribution of relaxation times and *ε*_∞_ is the dielectric permittivity at the limit of high frequencies. The spectral shape of the *α*-relaxation process can be well described by the HN function; however, a slower Arrhenius-like process related with the presence of additives was also observed at lower frequencies. Thus, a sum of two HN functions was used for the description of the active dielectric response of all samples. The relaxation times at maximum loss (*τ*_max_) are presented herein and have been analytically obtained by the Havriliak−Negami equation (Equation (3)):(3)$$\tau_{max,k}=\tau_{HN,k}\sin^{-1/m}\left(\frac{\pi m_{k}}{2(1+n_{k})}\right)\sin^{1/m}\left(\frac{\pi m_{k}n_{k}}{2(1+n_{k})}\right)$$

These relaxation times correspond to the α-relaxation process corresponding to the *T*_g_ dynamics as well as a slow Arrhenius-like process. At even lower frequencies, *ε*″ rises due to the conductivity (*ε*″ = *σ*/(*2πfε_f_*), where *σ* is the dc conductivity and *ε_f_* the permittivity of free space). The conductivity contribution has also been taken into account during the fitting process.

#### 2.4.5. Melt Flow Rate

The melt flow rate (MFR, g/10 min) of all compounds was measured at 230 °C and 2.160 kg, according to EN ISO 1133-1 standard [43], using a Dynisco model 4004 capillary rheometer (Dynisco Europe GmbH, Heilbronn, Germany).

#### 2.4.6. Mechanical Properties

Izod impact strength tests according to ISO180 [44] and tensile strength tests according to ISO527 [45] were determined for the developed FR compounds. For the determination of the mechanical properties, rectangular and dog-bone specimens from each compound were prepared by injection molding at 200–210 °C in an Arburg Allrounder 370C machine (Loßburg, Germany). Regarding the impact tests, 10 unnotched specimens of 80 × 10 × 4 mm^3^ per formulation were measured in an Instron Wolpert PW5 apparatus and the determined impact strength (*α*_iu_, in kJ/m^2^) was calculated according to Equation 4, where *E* is the absorbed energy during impact in J, *h* is the width and *b* is the thickness of each specimen in mm.
(4)$$a_{\mathrm{iu}}=\frac{E}{h\times b}\times 10^{3}$$

On the other hand, the impact behavior of the produced corrugated conduits was determined according to the EN IEC 61386-22:2021 standard (Resistance to impact) [2]. The acceptable impact value for medium type conduits is 2 J. Regarding the tensile tests, 10 dog-bone specimens per formulation of 6 × 2.2 mm^2^ at the neck and a gauge length of 50 mm were measured in an Instron 4416 apparatus. From the received stress–strain curves, the stress at yield (tensile strength, *σ*_max_), strain at break (*ε*_max_), and Young’s Modulus (*E*) were determined.

## 3. Results & Discussion

### 3.1. Lab-Scale Compounds Development and Evaluation of FR Performance

Two FR compounds were developed (FR1, FR2, Table 1) as candidates for the manufacture of flame-retarded and ageing-resistant corrugated conduits. First of all, the thermal stability of the pure FR additives was studied (Figure 1a) in order to correlate the anticipated FR mechanism (gas and/or condensed phase) to additive type and then to formulated compounds. The cyclic phosphonate ester (ADD1) showed a single mass-loss step beginning at ca. 259 °C (*T*_5%_), reaching a maximum decomposition rate at ca. 335 °C (*T*_d_), and leaving a residue of ca. 9 wt.% at 600 °C, thus exhibiting a minor charring ability [30,31,32]. On the contrary, the NOR-HAS compound (ADD2) shows a two-step decomposition curve, with the first step having its maximum mass-loss rate at ca. 277 °C (*T*_d1_), corresponding to the cleavage of the N-O-R bond, and exhibiting a weight loss of ca. 25 wt.%, and the second step, which is the decomposition of the carbon lattice of the additive, occurring at ca. 426 °C (*T*_d2_) [27]. The residue left at 600 °C was only 2 wt.%, thus proving that the pertinent additive is released mainly in the gas phase. Regarding the phosphinates (ADD3, ADD4), they decompose above 320 °C and release gases such as PH_3_ (*T*_d2_ = 329 °C) and H_2_O (*T*_d3_ = 400 °C), meanwhile leaving aluminum pyrophosphate as a heavy inorganic char [33,34,35,36] of ca. 75 wt.% at 600 °C, rendering these additives as good candidates to act in the condensed phase when added at high contents (e.g., 20 wt.%); this stable thick layer would block oxygen penetration and/or flame propagation to the polymer matrix. Moreover, ADD3 shows a three-step decomposition since, apart from the phosphinate fraction, which is the main ingredient of the pertinent additive, it contains also a NOR-HAS-type adjuvant. Therefore, the first step (*T*_d1_), occurring at ca. 292 °C, corresponds to the decomposition of the NOR-HAS adjuvant, which also lies within the decomposition range of the pure NOR-HAS additive (ADD2). On the contrary, ADD4, which is cited as a hypophosphite adjuvant, contains only mixtures of phosphinates, thus only two decomposition steps are clearly observed, with the step corresponding to water (300–400 °C) formation being more intense.

Regarding flame retardancy, FR1 compound (ADD1, ADD2, 11 wt.%) is anticipated to operate simultaneously in both the gas and condensed phase. The phosphonate (ADD1) and the NOR-HAS (ADD2), upon heating (high temperature, thermolysis), release phosphorous- (PO***^·^***), nitrogen- (NO***^·^***, N***^·^***), and alkyl-type (RO***^·^***, R***^·^***) radicals, which are very reactive and, on one hand, they cause rapid degradation of PP in the solid phase, leading to carbonization, while on the other hand, in the gas phase, they quench active radicals from the flame zone (O***^·^***, OH***^·^***, CH_3_***^·^***), thus terminating the fire [27,29,30]. The latter was verified by TGA (Figure 1b), where a ca. 75 °C lower *T*_5%_ value vs. reference PP was recorded (Appendix A), underlining the rapid degradation of PP in the presence of NOR-HAS and the gas phase side of the FR mechanism. On the other hand, the condensed phase part can be correlated to the *T*_d_ increase (by ca. 14 °C), with no significant residue increase, since the sample ended up at a residue of ca. 2 wt.%, similar to the PP value. This TGA-based FR1 function in the gas and condensed phase was macroscopically confirmed during the UL94V test: the specimens, after ignition, immediately extinguished, presenting a low char formation in their bottom edge with a simultaneous dripping that did not cause ignition of the cotton indicator (non-flaming dripping). The dripping and the gaseous byproducts resulted in an average mass loss of ca. 9.1 wt.%. The intense radical decomposition of PP in the solid phase is mainly induced by the thermolysis of the NOR-HAS component, resulting in chain scission and promoting non-flaming dripping, as observed by the increased melt flow [27,36]. V0 classification was reached for FR1, with a total burning time recorded at 13.7 s, while a very low smoke formation was evidenced, thus rendering the particular compound as promising for developing Halogen Free and Low Smoke (HFLS) conduits. The latter feature will be quantifiably evaluated in the following Smoke Density tests in samples of corrugated conduits, according to EN IEC 61034-2 standard [4].

On the other hand, in FR2, the flame retardancy mechanism is anticipated to operate mainly in the gas phase. In our case, the phosphinates (ADD3 and ADD4) are applied at a total loading of only 4 wt.%; thus, charring is not anticipated to be the dominant mechanism. Apart from the produced PH_3_ and H_2_O gases, which cool and dilute the flame zone, bromine radicals (Br***^·^***) are also released in the gas phase by the decomposition of the phosphorous bromine salt contained in ADD3, which in turn quenches the radicals that enhance flame propagation [34,35,36,37,38]. TGA verifies the latter, since a reduction of the *T*_5%_ by ca. 40 °C was evidenced (Figure 1b, Appendix A) due to the formed gases and radicals. On the other hand, *T*_d_ and final residue were found to be similar to reference PP. The absence of FR condensed phase mechanism can also be correlated to the UL94V test, where FR2 samples ignited faster and intense flaming dripping was directly observed. The total burning time was determined at higher levels, i.e., 25.4 s, compared to FR1, which is laid within the V0 specifications; however, the ignition of the cotton indicator directly classifies FR2 in V2 category. In addition, a much higher average mass loss was determined, i.e., ca. 29.2 wt.%, reflecting the predominance of the gas phase mechanism, which in turn yielded intense flaming dripping. It is worthwhile mentioning that FR2, due to ADD3, which is a halogenated additive, contains ca. 980 ppm of bromine. Nevertheless, it complies with the current low-halogen standard (EN50642, [5]) for cable protection systems, which permits bromine content up to 1500 ppm and also complies with the stricter DINVDE V 0604-2-100 [46], which permits halogen content up to 1000 ppm. Furthermore, the smoke emissions during the UL94 test were qualitatively observed to be slightly more intense compared to FR1, which is expected since more gases are produced. However, the FR2 compound can still be promising for the manufacture of HFLS conduits and will also be checked in the following Smoke Density tests in samples of corrugated conduits, according to EN IEC 61034-2 standard [4].

### 3.2. Characterization of Lab-Scale Compounds

The FRs influence on PP thermal properties was evaluated by typical DSC analysis under nitrogen atmosphere (Figure 2a,b, Appendix A). The phosphonate (ADD1) and NOR-HAS presence at a loading of 11 wt.% in FR1 caused slower melt crystallization rates, i.e., *T*_c_ was found to be significantly decreased by ca. 7 °C, inducing higher mass fraction crystallinity (*X*_c_ = 36%) and *T*_m_^2^ was increased by 3 °C compared to PP. In the case of FR2, thermal properties more similar to PP were determined, i.e., a ca. 0.5 °C increase in *T*_c_ and *T*_m_, and an identical crystallinity of 32%, as a consequence of the lower loading of 4 wt.% of the phosphinates ADD3 and ADD4. Moreover, a broad shoulder on the melting endotherm of FR2 is observed, possibly attributed to the melting of smaller crystals. Overall, it seems that the effect of the FR additives’ incorporation on the thermal properties is more pronounced in FR1 compared to FR2.

The NOR-HAS additive (ADD2, component of ADD3) offers a versatile functionality, acting as a stabilizer against oxidative degradation at low temperatures (<150 °C) or as a radical generator at high temperatures. This behavior is attributed to the production of nitroxyl-type radicals (NO***^·^***), caused by the thermolysis of the nitroxylether, that quench the formed peroxyl radicals (ROO***^·^***) during the oxidation cycle [20,24,26,30,47,48]. The latter was evaluated in the FR compounds through the determination of Onset Oxidation Temperature (OOT) (Figure 2c) [30,49]. Accordingly, the OOT of reference PP was determined at ca. 246 °C, while it was found significantly increased by ca. 11 °C for FR1 (1 wt.% ADD2, Appendix A). Therefore, for FR1, oxidation resistance is anticipated. As can be seen from the OOT-DSC curve of FR1 (Figure 2c), there is a small endotherm at ca. 245 °C, which corresponds to the melting point of the phosphonate (ADD1, Appendix A). Ιn the case of FR2, OOT was not significantly increased (only by ca. 3 °C), obviously due to the lower amount of NOR-HAS component: the total FRs loading 4 wt.% in FR2 is lower compared to 11 wt.% in FR1, while the precise NOR-HAS content in ADD3 is not revealed by the manufacturer, but can be assumed to be lower than 1 wt.% in the final FR2 compound.

Since the FR compounds are intended for the manufacture of cable conduits, the influence of the FRs presence on the dielectric behavior of PP was also assessed. The results from BDS for FR1, FR2, and reference PP are presented in Figure 3, in the form of isothermal curves of the dielectric permittivity (*ε*′), dielectric losses (log*ε*″) over frequency (*f*), for all temperatures examined. As can be seen, no relaxation processes can be observed in the dielectric spectra for all systems. This is expected for reference PP due to the non-polar segments of the polymeric chains, leading to extremely low dielectric losses, near the experimental limit of BDS (*ε*″~10^−3^). The dielectric behavior of FR1 and FR2, is similar with that of reference PP, exhibiting low dielectric loss values (*ε*″~10^−3^) and dielectric permittivity values of *ε*′~2.5 and *ε*′~2.9, respectively (*ε*′_PP_~2.7). This result can be related with the relatively small percentage of additives (<12 wt.%) that is contained in FR1 and FR2. In the case of FR1, an increase in dielectric losses can be observed at low frequencies (*f* < 10^2^ Hz), which can be related to dc-conductivity (*σ*_dc_) or the EPE phenomenon. Finally, for all systems, the extremely low values of the dielectric losses could not permit the analysis of the dielectric spectra with the use of HN equations (Equations (3) and (4)).

The compounds’ processability was compared to reference PP (MFR = 1.34 g/10 min) via MFR measurements. The 11 wt.% content of ADD1 and NOR-HAS resulted in an MFR increase up to 2.16 g/10 min, which can be attributed to the simultaneous melting of ADD1 (*T*_m_ = 230–245 °C) and ADD2 (*T*_m_~120 °C), as shown in Appendix A. In FR2, no significant change on the MFR was observed, since the contained additives are infusible.

Regarding the mechanical performance of the FR compounds, which is also a critical property for the application of corrugated conduits, tensile and impact properties were studied and compared to the PP reference (Appendix A). Mechanical testing is strongly affected by the additives’ compatibility with the polymer matrix and the additives’ content as well as by the degree of homogeneity achieved during compounding. In general, the relative standard deviations (RSD, %) for the impact strength measurements were found to be higher for the compounds (ca. 14–21%) vs. reference PP (5.6%), indicating poor additives dispersion, especially for FR1. Moreover, all PP specimens remained unbroken during the impact test, which was expected, since they were unnotched; on the contrary, all FR1 specimens completely broke. Based on the melting characteristics of FR1 additives (Appendix A), both the cyclic phosphonate ester (ADD1) and the NOR-HAS (ADD2) might have partially or even totally melted during the extrusion and/or subsequent injection molding, thus creating different phases in the material bulk, which in turn could result in weaker spots. For the case of FR2, where the phosphinates (ADD3 and ADD4) are completely infusible additives, compounding permitted a more homogenous dispersion of the solid particles and distribution in PP matrix. During the impact tests, all specimens of FR2 remained unbroken, thus showing a similar behavior to PP. Regarding impact strength values (Appendix A), in FR1, where the FR loading is 11 wt.%, an increased *a*_iu_ compared to reference PP was determined, unlike in the case of FR2, where the low loading of 4 wt.% did not seem to significantly affect the *a*_iu_. In terms of tensile properties, the FR compounds at both total loadings (11 and 4 wt.%) exhibited similar values for Young’s modulus and stress at yield (tensile strength) with reference PP, but they presented a more brittle and stiffer behavior based on the strong decrease in the strain at break (*ε*_max_), with RSD values between 10% and 20% [19,48].

### 3.3. Performance of Lab-Scale FR Compounds during Ageing

A very important aspect for long-term and durable application, such as in the case of conduits, is the retention of both flame retardancy and mechanical properties. Therefore, the developed FR compounds were subjected to separate accelerated artificial ageing experiments: one accelerated heat ageing test, performed in an air circulating oven at 110 °C for 2000 h; and one accelerated weathering test in the presence of UV radiation and humidity also for 2000 h, in a climate chamber. Regarding flame retardancy, the two FR compounds maintained their initial (prior to ageing) UL94 classification, in both sets of accelerated ageing tests. Beginning with heat ageing (Figure 4a), FR1 was the best performing compound, with all the specimens constantly at V0 class and an excellent retention of the total burning time at ca. 14 s. FR2 also successfully remained in the V2 category but exhibited ups and downs in total burning time. Similar were the results after the UV weathering test (Figure 4b), with FR1 constantly in the V0 class, with insignificant variations (up to ca. 3 s) in total burning time; and FR2 constantly in the V2 class, with more distinct ups and downs in terms of total burning time. However, at the sampling intervals 1000, 1500, and 2000 h of weathering, all the specimens of FR1 exhibited increased stickiness after their removal of the climate chamber; however, this did not affect their performance in the UL94 test. This can be explained by the interaction of humidity with ADD1 at the temperature of 50–60 °C at which UV ageing took place, causing partial hydrolysis of the additive and thus showing a much lower melting point [31,50]. Finally, especially for the UV-aged samples, the UL94 bars were weighed prior to and after the UL94 test, so as to determine the mass that was lost through the formation of gases and dripping (Figure 4c). Accordingly, FR1 maintained the low (<10 wt.%) average mass loss during the flame test (although a mildly decreasing trend was observed), indicative for the gas and condensed-phase FR mechanism. FR2 also, in terms of mass loss during the UL94 test, exhibited a similar trend of ups and downs with total burning time (ca. 22–33 wt.%) as a consequence of the intense flame dripping. All aged specimens exhibited a macroscopically small increase in the smoke emission during the UL94 test, probably due to some minor ageing of the FR additives (as proved by the retention of UL94 classification); nevertheless, the emission levels are still considered very low. Summarizing, the developed FR compounds ideally retained their flammability against ageing under our experimental conditions; thus, they are considered as good candidates for upscaling to an industrial level.

Turning to the mechanical performance during ageing, first, we examined the retention of tensile strength in the heat ageing experiments in order to assess any thermo-oxidative stabilization effect of the FR additives. Typically, polyolefins with increasing ageing time exhibit oxidative degradation triggered by heat and/or UV radiation in the presence of oxygen [15,39,40]. In terms of tensile properties, PP, upon ageing, typically exhibits an increase in the Young’s modulus (*E*), with a strong reduction in the elongation at break, a behavior well-known as embrittlement [19,48,51,52,53,54]. This was observed clearly for reference PP (Figure 5). Therefore, a tolerance level of ±20% in the change of all three tensile values was arbitrarily selected so as to evaluate the mechanical performance of the FR compounds against heat ageing. Accordingly, the two FR compounds exhibited a more slowly increasing trend of tensile strength (*σ*_max_, Figure 5a), ending up at 2000 h at a total deviation in the range of 17–19%. Especially for the initial 500 h, the relative increase in tensile strength can be due to annealing of the polymer [19]. Regarding the strain at break (*ε*_max_, Figure 5b) and Young’s modulus (Figure 5c), no clear trend was observed, as anticipated, due to embrittlement: FR1 was quite stable throughout the ageing time, ending up with an insignificant reduction of *ε*_max_ in the range of ca. 7% and a reduction of 10% for *E* at 2000 h. FR2, surprisingly, exhibited an increasing trend of *ε*_max_ from as soon as 500 h, reaching 20% change and a constant increase in *E* even reaching 47%. It can be concluded that the developed FR compounds exhibited embrittlement during heat ageing, but at much lower extent than reference PP.

Turning to UV weathering, Izod impact strength was monitored (Figure 6a) in order to simulate the maintenance of PP conduits performance for outdoor applications, since impact is the most critical mechanical property for such products. Already, reference PP exhibited a stable behavior of impact strength during the weathering process [19,47,51,52,53,54], with all measured specimens again not being broken during the impact test. On the contrary, FR1, at 500 h, exhibited a strong decrease of ca. 44%, which, at 1000 and 1500 h, increased again close to the initial value and at ended up at 2000 h with a 50% total decrease. The standard deviation values are also high (18–65.5%) vs. ageing time, which is again an indication of poor additives dispersion or phase separation in the compounded material (FR1). In addition, all FR1 specimens broke during the impact tests. Therefore, it is not easy to accurately determine the effect of ageing on impact strength; nevertheless, it is obvious that FR1 shows impact strength deterioration, probably due to ADD1 partial hydrolysis as mentioned above. In contrast to FR1, FR2 exhibits a very similar behavior to reference PP (all aged specimens remained unbroken during the test) with impact strength maintenance throughout the UV weathering time. Moreover, impact specimens were weighed prior to and after each ageing interval and the mass gain due to humidity (water uptake, Figure 6b) was calculated. Accordingly, FR2 exhibited very low water uptake, similar to PP, as expected, since PP is not considered as hygroscopic material. On the contrary, FR1 showed a much higher water uptake, which increased with UV-ageing time, proving the interaction of ADD1 with humidity and indirectly verifying the partial hydrolysis that the pertinent additive suffered.

In order to gain deeper insight regarding the UV-ageing process, FT-IR and MFR were also monitored. FT-IR ATR analysis evaluates the surface of the material, since the IR beam penetrates only a few μm in the sample, while MFR provides information about the bulk material. Regarding FT-IR analysis during UV ageing (Figure 7a–c), FR1 at 500 h shows a peak at ca. 1700 cm^−1^, which is within the carbonyl range, and another one at ca. 3300 cm^−1^, which is within the hydroxyl region [41,42]. Both peaks exhibit increasing intensity with ageing time and suggest ageing byproducts. Moreover, significant changes in the spectra of the UV-aged FR1 samples (Figure 7a) are noticed in the area of 900–1400 cm^−1^, which can be explained by the hydrolysis that ADD1 suffered during interaction with the humidity of the climate chamber, verified macroscopically by the observed stickiness and impact strength deterioration. On the contrary, FR2 shows no change in the spectra during ageing (Figure 7b), underlining the stability that the pertinent FR system provided. Regarding reference PP (Figure 7c), very weak peaks only in the carbonyl range after 500 h are observed, verifying that a minor photo-degradation occurred.

MFR was also measured at the sampling intervals of 1000 and 2000 h of UV ageing. Accordingly (Appendix A), reference PP and FR2 show a remarkable MFR stability, while FR1 exhibits a ca. sevenfold increase in MFR already from 1000 h. This is in line with the observation of carbonyl peaks in the FT-IR spectra of the FR1 samples from UV ageing (Figure 7a). This rapid increase in MFR of FR1 aged samples could be explained by the hydrolysis of ADD1, which results in much lower melting point species, thus strongly affecting the MFR and/or chain-scission of PP [19,54]. The latter verifies that FR1 formulation requires the addition of an acid scavenger, like zinc stearate or antioxidants, so as to inhibit the observed degradation of the ADD1 and the PP matrix [48,50].

The BDS results for FR1 and FR2 after heat ageing at 110 °C for 0, 1000 and 2000 h are presented in Figure 8, in the form of isothermal curves of the dielectric permittivity (*ε*′) and dielectric losses (log*ε*″) over frequency (*f*), for all temperatures examined. As was mentioned before, in the absence of thermal ageing (0 h), FR1 and FR2 show extremely low values of dielectric losses (log*ε*″); thus, their dielectric behavior cannot be further discussed. In the case of FR1, thermal ageing leads to major changes in the dielectric spectra (Figure 8a,b). After thermal treatment for 1000 h (40 days) at 110 °C, dielectric loss values increase at all examined temperatures (*ε*″~10^−2^) and two relaxation processes can be observed in the dielectric loss spectra (Figure 8a). The fast one corresponds to *α* process, related to *T*_g_, and the slower one corresponds to an Arrhenius-like process related with the presence of additives in the polymeric matrix of PP. After thermal treatment for 2000 h (80 days), the two processes become more distinct, with the dielectric losses increasing to even higher values (*ε*″~10^−1^). These changes can also be observed at the dielectric permittivity curves (Figure 8b), where there is a gradual increase of dielectric permittivity (*ε*′) with increasing ageing time, for all temperatures examined, with a value of *ε*′~3.4 for 1000 h and *ε*′~3.6 for 2000 h. On the other hand, in the case of FR2 (Figure 8c,d), thermal ageing for 1000 and 2000 h does not have any significant effect on the dielectric spectra. Dielectric loss values for both ageing times remain very low at all temperatures studied, making the analysis of the dielectric loss spectra using the HN equations meaningless. For that reason, in Figure 8c,d, the dielectric response at only two indicative temperatures is shown. In terms of dielectric permittivity, a small drop can be observed (*ε*′~2.35) for both 40 and 80 days, at temperatures *T* = 273 K and *T* = 298 K.

Turning to BDS results after UV ageing (Figure 9), the results for FR1 after UV ageing for 0, 1000, and 2000 h are presented, in the form of isothermal curves of the dielectric permittivity (*ε*′) and dielectric losses (log*ε*″) over frequency (*f*), for all temperatures studied. In accordance with thermal ageing, UV ageing seems to also have a significant impact on the dielectric spectra. After treatment for 1000 h and 2000 h, one relaxation process with a broad distribution of relaxation times can be observed, whereas, in the absence of UV treatment, the dielectric spectra did not show any similar process. In terms of ageing duration effects, there is no significant difference between 1000 h and 2000 h. Dielectric loss values for both ageing times are increased (*ε*″~10^−1^) for all temperatures studied and the dielectric permittivity spectra show one distinct characteristic step, corresponding to the relaxation process, with a lower permittivity value of *ε*′~2.3. Due to absence of any effects to dielectric behavior of FR2 after heat aging, dielectric measurements were not performed on FR2 UV aged samples. The dielectric losses of FR2 always remain very low.

Analysis of dielectric data of FR1 after heat or UV ageing with different ageing times was employed with the use of HN equations on the dielectric permittivity data (log*ε*″). The relaxation times of the molecular processes obtained from this procedure were plotted over reciprocal temperature (1000/*T*) (Appendix A). In the case of FR1 without ageing, the dielectric spectra could not be analyzed employing the HN analysis. After thermal ageing, an Arrhenius-like process can be observed in the dielectric spectra at high temperatures for both ageing times (1000 and 2000 h), corresponding to molecular mobility of charges that exist in the polymeric matrix due to the additives. In addition, a relaxation process at lower temperatures can be observed corresponding to segmental *α* process, the molecular motion related to *T*_g_, with a characteristic VFT dependence. For *t* = 100 s, the extrapolated fit of the VFT curve predicts *T*_g_ values of −47.7 °C, for 1000 h, which is far below the *T*_g_ of PP from other studies. In the case of FR1, thermally aged for 2000 h, both relaxation processes are shifted towards larger relaxation times, indicating slower dynamics, characterized by a higher *T*_g_ value of −30.9 °C. Regarding the magnitude of the relaxation processes, in Appendix A, the normalized dielectric strength (*T*Δ*ε*) is plotted over reciprocal temperature. As can be observed, the dielectric strength of both the Arrhenius-like and *α* process increases immensely with the increase in ageing time during thermal treatment (red line), whereas no differences are observed during UV aging.

### 3.4. Upscaling FR Compounds to Industrial Level

FR1 and FR2 formulations set the basis for the production of industrial-scale masterbatches (MB) at additive loadings 40 and 44 wt.%, respectively (Table 1), i.e., 4 and 10 times the initial loading of the FR1 and FR2 compounds. The formulations of the industrial MBs were specially designed according to the expertise of our industrial compounder (PLASTIKA KRITIS) and the required MB additive range required by our end-user (KOUVIDIS SA). A high MFR homo-polymer PP (25 g/10 min) was selected as carrier, so as to facilitate the high loading of the additives during MB compounding.

The prepared masterbatches were initially characterized in terms of thermal properties in order to verify the additives’ presence and functionality based on lab-scale prior analysis. Accordingly (Appendix A), both MBs showed two-step mass loss TGA curves, which are much more pronounced and clearly separated in comparison to the respective FR compounds (Figure 1b), since the additives are incorporated at a much higher loading. According to Appendix A, *T*_5%_ values are up to 100 °C lower than the respective value of the PP carrier, but far below the extrusion processing window (190–220 °C); therefore, no degradation or consumption of FR additives of the MBs is anticipated during further blending with reference PP in extrusion. *T*_d1_ and *T*_d2_ values correspond to physico-chemical reactions of the FR additives, as already explained in Figure 1a [33,34,35,36]. Finally, regarding residue, MB1 shows a value of ca. 10 wt.%, again underlining the minor charring ability of ADD1 and ADD2, while in MB2, ca. 30 wt.% was recorded as a consequence of the high charring ability of the contained phosphinates [34].

From the received MBs, FR1 and FR2 were reproduced, but by blending with the appropriate amount of reference PP in the twin-screw extruder, resulting in FRMB1 and FRMB2 (Table 2). This was performed in order to verify the efficiency of the produced MBs. UL94 bars were prepared by compression molding and the UL94 test was repeated. For FRMB1, the V0 class was reached, with a total burning time for five specimens determined at 13.3 s and an average mass loss per specimen due to dripping at 9.2 wt.%, very close to the respective FR1 (13.7 s and 9.1 wt.%). Similarly, in the case of FRMB2, the V2 class was reached, with a total burning time at 29.8 s and mass loss due to dripping at 23.3 wt.%, again very close to the respective initial FR2 values (25.4 s and 29.2 wt.%, respectively). Moreover, impact specimens of FRMB1 and FRMB2 were prepared so as to evaluate the homogeneity of the compounds developed via the MBs. Indeed, as observed in Figure 10, the compounds developed from the MBs show slightly higher average impact strength, but, meanwhile, a much lower standard deviation, which verifies that the dilution from the MBs ensures a better and more homogenous dispersion of the FR additives. Indicatively, the RSD values for FRMB1 and FRMB2 are 2.7% and 4.9%, respectively, much lower than the ones given in Appendix A.

Furthermore, after securing the optimum FR dispersion with the use of the MBs, an attempt to produce FR compounds at the lowest possible MB concentration that would, meanwhile, result in similar FR performance was performed. This attempt aims mainly at a cost-effective approach for industrializing these compounds, but especially in the case of FR2, also at a reduced halogen content. Accordingly (Table 2), six new FR compounds (FRMB3-FRMB8) were developed based on MB1 and two (FRMB9, FRMB10) based on MB2, via twin-screw extrusion of the desired amounts of reference PP and the required MB.

Regarding compounds developed from MB1 (Table 2, Figure 11), FRMB3 and FRMB4, where the FR loading is 9 and 8 wt.%, respectively, similar total burning times are reached and V0 class is retained, since the dripping does not cause cotton ignition. Moreover, mass loss due to dripping is minimized for FRMB4 (8 wt.% FR loading). The latter constitutes a remarkable FR performance for the case of PP, since V0 at such low loading and for bulky applications like conduits is hardly found in the literature or in commercial additives. There are, however, novel FR systems, such as oxy-imide nitrogen based radical generators, not yet commercialized, that promise V0 class in PP only at 6 wt.% loading [55,56,57,58,59]. When reducing the FR loading further (FRMB5-FRMB7) up to 3.52 wt.%, an increase in total burning time is observed, which is still within the V0 specifications (<50 s for 5 specimens), but the most important observation is that dripping leads to cotton ignition, thus the V2 class is attained. Meanwhile, apart from the increase in total burning time, mass loss due to dripping is also significantly increased. Finally, FRMB8, which contains an extremely low FR loading of only 2.2 wt.%, FR performance is completely lost, with four out of five specimens tested failing completely in the UL94 test and being burnt up to the clamp, as indicated by the average mass loss of 89.6 wt.%. Therefore, the particular compound is non-classified (NC). Turning to the compounds developed from MB2 (FRMB9, FRMB10, Table 2), the FR loading is already very low in FR2 and FRMB2, i.e., 4 wt.%, and with V2 class already attained for those, the optimization margins are very narrow. Nevertheless, in FRMB9 (Figure 11), where the FR loading was reduced to 3.2 wt.%, mildly increased total burning times and mass loss due to dripping were determined. However, a significant reduction in the bromine content from 980 ppm in FR2 and FRMB2 to 784 ppm in FRMB9 was achieved, thus FRMB9 complies even with the strict DIN VDE V 0604-2-100 standard [46] that admits bromine up to 1000 ppm. Finally, a further reduction of the FR loading to 2 wt.% leads to a non-classified compound, i.e., FRMB10; however, two out of five samples were still categorized as V2.

From the lab-scale compounds and for cost-efficiency reasons, FRMB7 based on MB1 and FRMB9 based on MB2 were selected and industrial-scale production tests were performed according to these formulations, producing two (2) respective types of corrugated conduits (C1, C2) of ø20 mm outer nominal diameter. The produced conduits were characterized prior to and after 2000 h of accelerated UV ageing according to EN IEC 61386-22:2021 standard for their resistance to flame propagation and impact [2]. In addition, their smoke density was determined according to EN IEC 61034-2 standard [4] only prior to ageing (Table 3).

Accordingly, conduit C1/0h is considered as non-flame propagating, according to IEC 61386-22 [2], since self-extinguishing behavior was evidenced in all six measured samples, with burning times (average burning time 1.84 s) much lower than 30 s that the standard permits. Non-flame propagating behavior was maintained after 2000 h (C1/2000h) and, although the burning time was found increased, it is still far below the limit of 30 s. Similarly, both the C2/0h and C2/2000h conduit samples are considered as non-flame propagating. The latter verify the results from the lab-scale and the UL94 tests, and underline the stability of FRs performance against aging. Turning to impact resistance, the produced C1/0h and C2/0h conduits are considered as medium-type conduits, according to IEC 61386-22 standard [2], since they can endure impact energy higher than 2 J. The impact behavior of the conduits is fully in line with the lab-scale tests of the FR compounds, and, even though a decrease ranging from 16 to 25% was observed after 2000 h, samples C1/2000h and C2/2000h still retain the medium-type impact category. Last but not least, the smoke density of the C1/0h and C2/0h was determined, according to EN IEC 61034-2 standard [4]. In both cases, the light transmittance was much higher than the limit of the standard (60%), thus both conduits are truly low smoke (LS), verifying the macroscopical observation from the UL94 tests in the lab scale. The result, when considered alongside the totally halogen-free ADD1 and ADD2, or the designed halogen level at 784 ppm, is the creation of real halogen-free and low-smoke (HFLS) conduit products.

## 4. Conclusions

Two FR PP compounds were developed in the lab-scale aiming at the production of halogen free and low smoke (HFLS) conduits designated for cable protection. FR1 is completely halogen free (zero halogen) since it comprises two different additives: a cyclic phosphonate ester (ADD1) and a N-alkoxy hindered amine grade (ADD2) at a weight ratio of 10:1 and a total FR loading of 11 wt.%. On the other hand, FR2 consists of a mixture of hypophosphite-based additives (ADD3, ADD4) at loading of only 4 wt.%, with ADD3 also containing NOR-HAS and bromine adjuvants, which result in a final bromine content of 980 ppm; nonetheless, it can be considered as halogen free since it complies with the current EN50642 standard. FR1 reaches the V0 class and seems to operate in both the gas and the condensed phase, while, for FR2, V2 is reached and a typical gas phase mechanism with production of gases and bromine radicals follows. Apart from flame retardancy, the application of conduits requires satisfactory mechanical performance, which is ensured by the relatively low FR loading in both compounds aiming at mechanical properties close to reference PP. A further aspect of the conduit application is weathering resistance, which was also studied in terms of UV and heat ageing. Accordingly, both FR compounds retained their initial FR performance ideally throughout ageing tests. However, during UV weathering, FR1 suffered from degradation, probably due to the partial hydrolysis of ADD1 additive, as depicted from macroscopical observation of stickiness on the surface of the aged specimens, from carbonyl species in the FT-IR spectra and from a severe increase in the MFR of the aged products. BDS results obtained on thermally and UV-aged samples indicated a strong modification of the semicrystalline structure and enhanced molecular mobility, supporting the finding that FR1 compound suffers from degradation during aging. On the contrary, FR2 exhibited a very stable behavior in terms of mechanical properties, FT-IR, and MFR. The promising results of the FR1 and FR2 compounds prior to ageing led to the industrial development of the respective masterbatches MB1 and MB2 in a total loading of 44 and 40 wt.%, respectively. From the developed MBs, not only the successful reproduction of the initial was achieved, but also new compounds of reduced FR loading were developed. Accordingly, from MB1, the developed zero-halogen compound containing only 8 wt.% of ADD1 and ADD2 resulted in V0 classification, which constitutes a breakthrough for bulky PP applications. Meanwhile, the compound containing only 3.2 wt.% of ADD3 and ADD4 resulted in the V2 class, but most importantly, it shows a strongly reduced bromine content of 784 ppm compared to FR2. From the lab-scale compounds, two formulations (FRMB7 and FRMB9) were reproduced in the industrial scale via masterbatch preparation, and two types of corrugated conduits of 20 mm outer nominal diameter were produced. The developed conduits are considered as non-flame propagating and medium-impact type, according to EN IEC 61386-22:2021 standard. These key properties were highly retained after 2000 h of accelerated UV ageing performed directly on conduit samples. Last but not least, both grades of the developed conduits are considered as low smoke according to EN IEC 61034-2 standard. The overall work described, from the lab-scale tests up to the real industrial production scale of conduits, sets the basis for HFLS conduits based on PP that come as viable and environmentally friendly alternatives to the already commercially available PVC products.

## Figures and Tables

**Figure 1 polymers-16-01298-f001:**
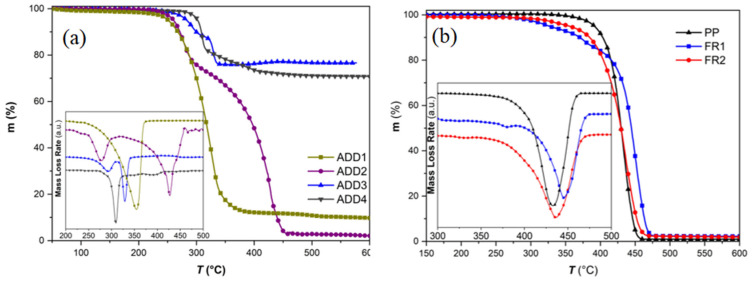
TGA curves of (**a**) the raw FR additives and (**b**) the developed compounds vs. reference PP.

**Figure 2 polymers-16-01298-f002:**
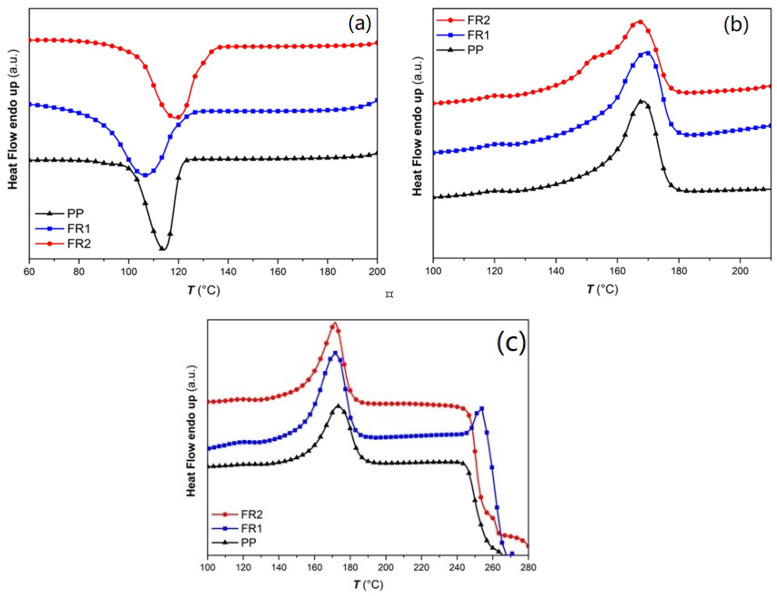
(**a**) Cooling curves, (**b**) 2nd heating DSC curves of FR compounds vs. reference PP (nitrogen atmosphere), (**c**) Determination of Oxidation Onset Temperature (OOT) under air atmosphere.

**Figure 3 polymers-16-01298-f003:**
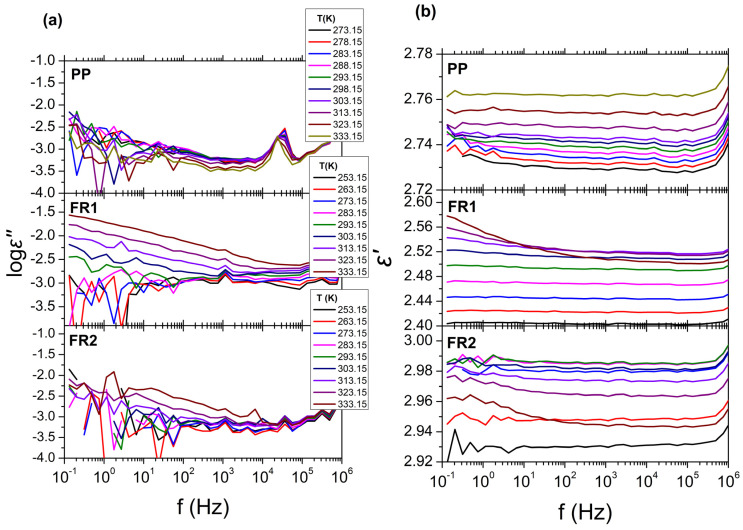
BDS isothermal curves for all temperatures examined with (**a**) the dielectric losses (log*ε*″) and (**b**) the real part of permittivity (*ε*′) over frequency (*f*). From top to bottom, the isothermal curves refer to: reference PP, FR1, and FR2.

**Figure 4 polymers-16-01298-f004:**
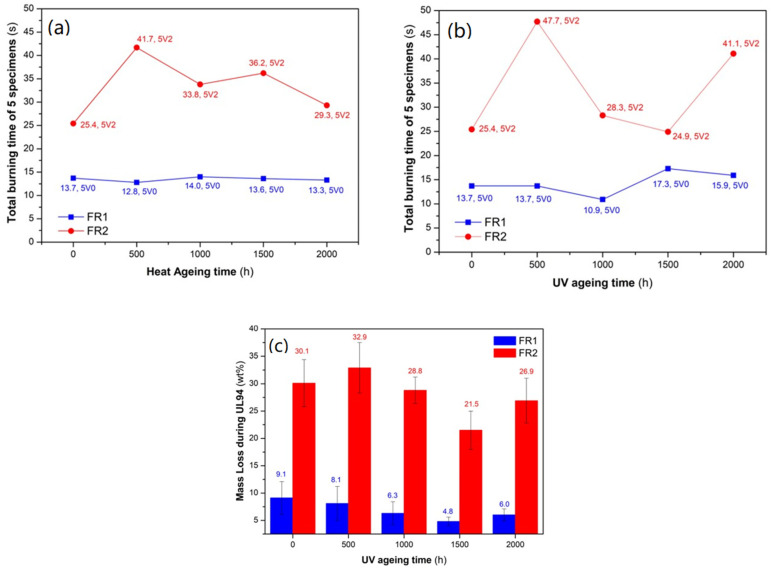
Total burning time for 5 specimens during artificial ageing experiments. (**a**) Heat ageing at 110 °C for 2000 h, (**b**) UV weathering for 2000 h, (**c**) Average mass loss during UL94 test for the UV aged samples.

**Figure 5 polymers-16-01298-f005:**
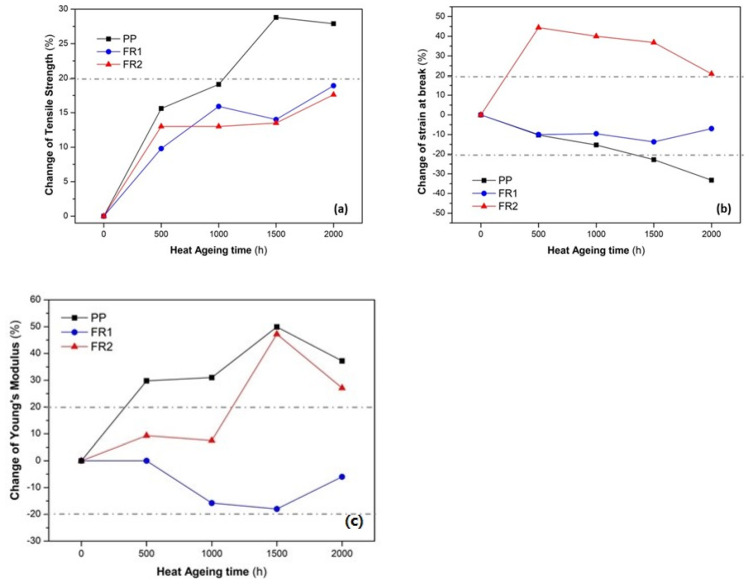
Evolution of tensile properties of FR compounds during heat ageing. (**a**) tensile strength, (**b**) strain at break, (**c**) Young’s Modulus.

**Figure 6 polymers-16-01298-f006:**
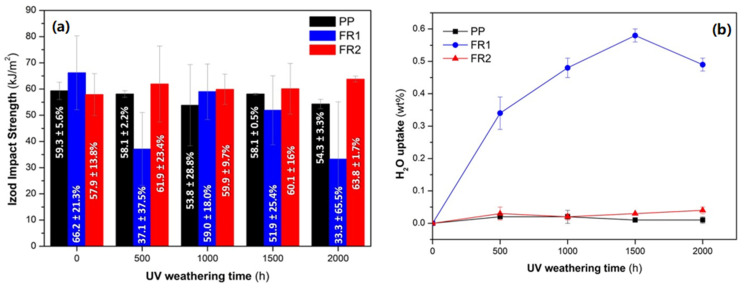
(**a**) Evolution of Izod impact strength and (**b**) water uptake of FR compounds during UV weathering.

**Figure 7 polymers-16-01298-f007:**
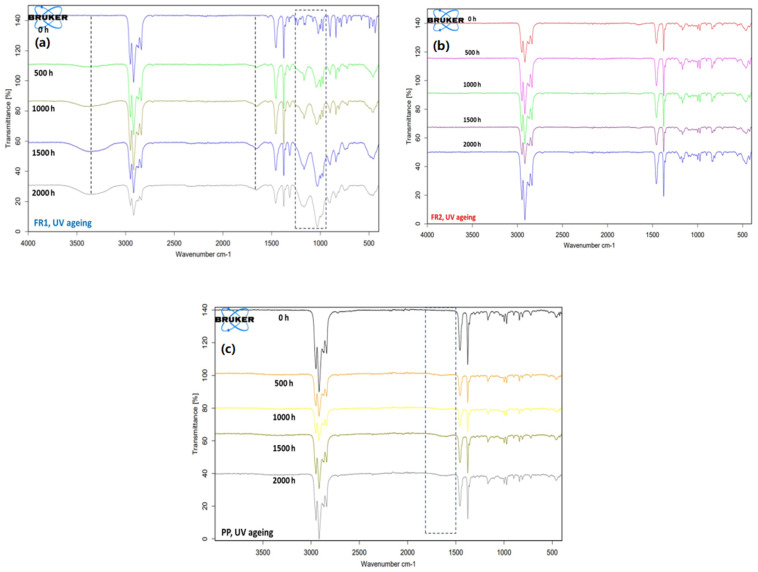
FT-IR ATR spectra of FR compounds (**a**,**b**) and reference PP (**c**) during UV ageing. Dashed lines highlight the changes in the spectra as a consequence of ageing.

**Figure 8 polymers-16-01298-f008:**
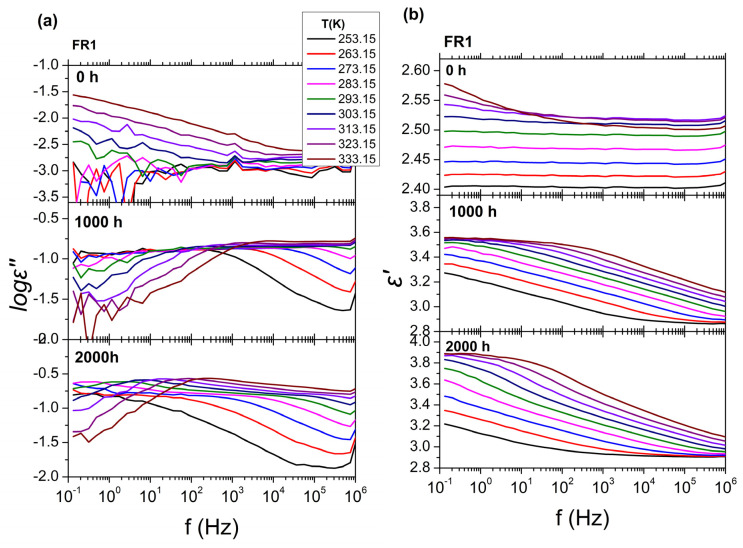
BDS isothermal curves of FR1 and FR2 after thermal ageing at 110 °C, for all temperatures examined, with (**a**,**c**) the dielectric losses (log*ε*″) and (**b**,**d**) the real part of permittivity (*ε*′) over frequency (*f*). From top to bottom for each graph, the isothermal curves refer to different ageing times: 0 h, 1000 h, and 2000 h.

**Figure 9 polymers-16-01298-f009:**
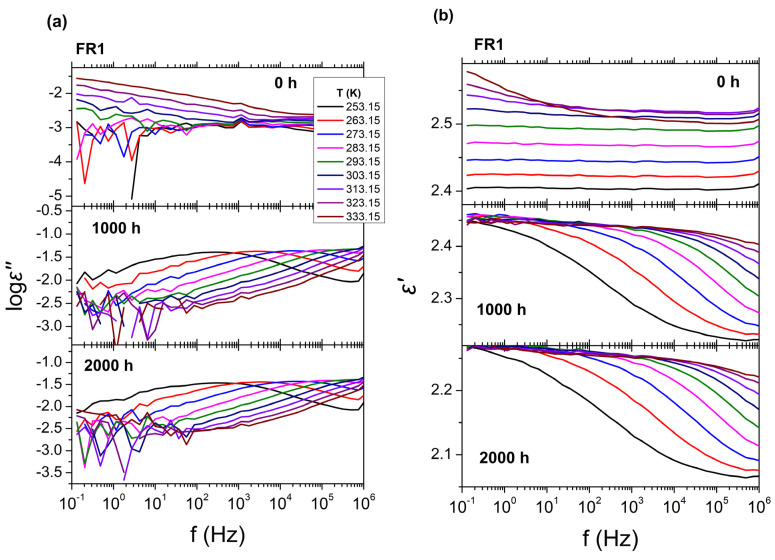
BDS isothermal curves of FR1 after UV ageing for all temperatures examined, with (**a**), the dielectric losses (log*ε*″) and (**b**)the real part of permittivity (*ε*′) over frequency (*f*). From top to bottom for each graph, the isothermal curves refer to different ageing times: 0 h, 1000 h, and 2000 h.

**Figure 10 polymers-16-01298-f010:**
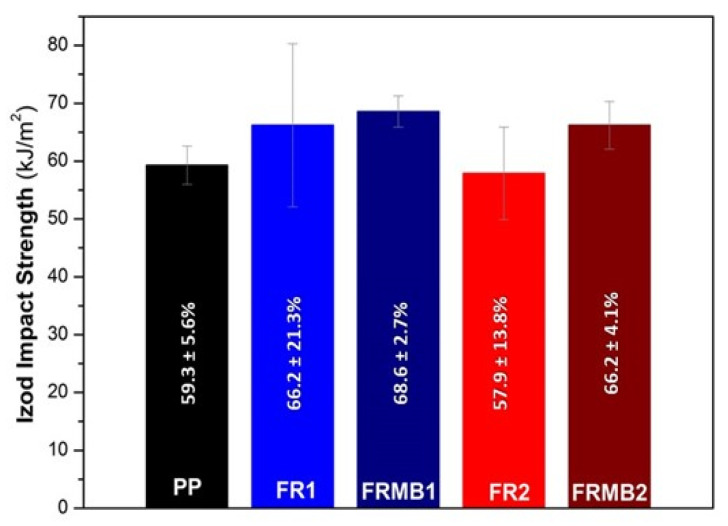
Izod impact strength of FR1, FR2 compounds vs. FRMB1, FRMB2 compounds.

**Figure 11 polymers-16-01298-f011:**
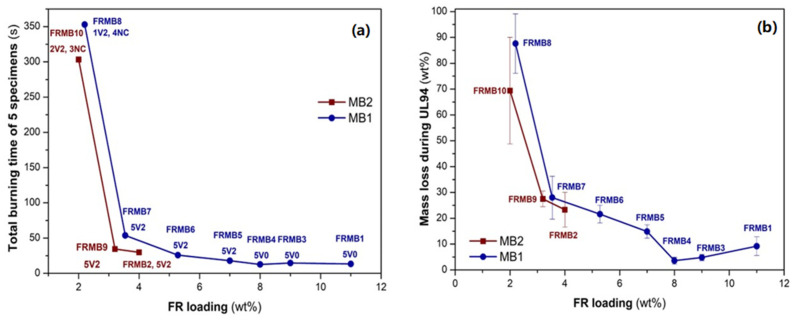
(**a**) Total burning times as a function of FR loading for different formulations developed from the industrial masterbatches (MB1, MB2) (**b**) Average mass loss per specimen during dripping in the UL94 test.

**Table 1 polymers-16-01298-t001:** Composition of the developed formulations in wt.%. Lab-scale compounds (FR1, FR2) and industrial scale masterbatches (MB1, MB2).

Formulations	PP	PP Carrier	ADD1	ADD2	ADD3	ADD4	Bromine	Total
	[wt.%]	[wt.%]	[wt.%]	[wt.%]	[wt.%]	[wt.%]	[ppm]	[wt.%]
*Lab-Scale Compounds*								
FR1	89	-	10	1	-	-	-	11
FR2	96	-			3.5	0.5	980	4
*Industrial Scale Masterbatches (MBs)*								
MB1	-	56	40	4	-	-	-	44
MB2	-	60	-	-	35	5	9800	40

ADD1: Aflammit PC900, ADD2: Flamestab NOR116, ADD3: Phoslite B713A, ADD4: Phoslite B85AX. Bromine content in ppm calculated according to the nominal ADD3 content of 2.8% given by the manufacturer.

**Table 2 polymers-16-01298-t002:** Lab-scale compounds developed from MBs at several dilution rates in wt%. FRMB1, FRMB3-FRMB8 developed from MB1, FRMB2, FRMB9-FRMB10 developed from MB2.

Formulations	PP	MB1	MB2	Bromine *	Total **
	[wt.%]	[wt.%]	[wt.%]	[ppm]	[wt.%]
FRMB1	75	25	-	-	11
FRΜΒ3	79.5	20.5	-	-	9
FRΜΒ4	81.8	18.2	-	-	8
FRΜΒ5	84	16	-	-	7
FRΜΒ6	88	12	-	-	5.28
FRΜΒ7	92	8	-	-	3.52
FRΜΒ8	95	5	-	-	2.2
FRMB2	90	-	10	980	4
FRΜΒ9	92	-	8	784	3.2
FRΜΒ10	95	-	5	490	2

* Bromine content calculated according to the nominal ADD3 content of 2.8% given by the manufacturer. ** Total concentration of FR additives in wt.% as calculated from the MB amount.

**Table 3 polymers-16-01298-t003:** Key properties of produced PP pliable corrugated conduits.

Conduit Types	MB Type	MB Dosing (%)	Resistance to Flame Propagation (s)	Resistance to Impact (J)	Smoke Density (%)
C1/0h	MB1	8	1.84	2.5	72.8
C1/2000h			3.36	2.0	*n.d.*
C2/0h	MB2	8	2.72	3.0	93.8
C2/2000h			2.56	2.5	*n.d.*

*n.d.*: not determined.

## Data Availability

The original contributions presented in the study are included in the article/Appendix A, further inquiries can be directed to the corresponding authors.

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
