# Peer review of "Flame-Retarded and Heat-Resistant PP Compounds for Halogen-Free Low-Smoke Cable Protection Pipes (HFLS Conduits)"

_polymers, 2024, doi:10.3390/polym16091298_

Round 1

Reviewer 1 Report

Comments and Suggestions for Authors

Dear Editor,

In this study, Authors aimed to develop appropriate halogen free PP compounds for the manufacture of conduits. Two FR systems were examined; one comprising a cyclic phosphonate ester and a monomeric N-alkoxy hindered amine synergist, and a second one comprising hypophosphite moities along with two synergists, a N-alkoxy piperidine (NOR-HAS) and a phosphorous-bromine salt. Thermomechanical, dielectric properties of the end products was examined. The work is well prepared and well constructed. Before the publishing following points should be taken into consideration.

Comment 1. First of all, Abstract is too long. It would be better if it could be shortened

Comment 2. Composition for the heterophasic PP copolymer should be given. Table 1 is confusing. What is the definition for the “Lab scale compounds derived from MBs”. How is the formulations developed under this subject? For example, what is the difference between FR1 and FRMB1? It is reported in the text” Accordingly (Table 1), six new FR compounds (FRMB3-FRMB8) were developed based on MB1 and two (FRMB9, FRMB10) based on MB2, via twin-screw extrusion of the desired amounts of reference PP and the required MB” however Table 1 itself is confusing. Why is there such a high amount of additives in industrial scale MBs? Description for ADD1, ADD2, ADD3, ADD4 and Bromine content analysis should be given under the Table as a footnotes. The chemical structures of these compounds can also be given in supporting info.

Comment 3. Figure 3. inset data is not read well.

Comment 4. An evaluation was made about additives compatibility with the polymer matrix between line and their effects on mechanical properties were examined. Similarly, was there a deviation in the test samples in the FR tests?

Comment 5. Figure 8 and Figure 9: inset data is not read well.

Author Response

Comment 1: First of all, Abstract is too long. It would be better if it could be shortened.

Response 1: Abstract was modified and shortened as follows so as to be at ca. 200 words. New abstract was added in the text and was highlighted in yellow (Lines 19-35).

Conduits are plastic tubes extensively used to safeguard electrical cables, traditionally made from PVC. Recent safety guidelines seek alternatives due to PVC's emission of thick smoke and toxic gases upon fire incidents. Polypropylene (PP) is emerging as a viable alternative but requires modification with suitable halogen-free additives to attain flame retardancy (FR) while maintaining high mechanical strength and weathering resistance, especially for outdoor applications. The objective of this study was to develop two FR systems for PP: one comprising a cyclic phosphonate ester and a monomeric N-alkoxy hindered amine synergist achieving V0, and another with hypophosphite and bromine moieties, along with a NOR-HAS synergist achieving V2. FR performance along with mechanical properties, physicochemical characterization and dielectric behavior were evaluated prior to and after 2000 h of UV weathering or heat ageing. The developed FR systems set the basis herein for the production of industrial-scale masterbatches, from which further optimization to minimize FR content was performed via melt mixing with PP towards industrialization of a low-cost FR formulation. Accordingly, two types of corrugated conduits (ø20 mm) were manufactured; their performance in terms of flame propagation, impact resistance, smoke density and accelerated UV weathering stability classified them as Halogen Free Low Smoke (HFLS) conduits, meanwhile meeting EU conduit standards without significantly impacting conduit properties or industrial processing efficiency.

Comment 2. Composition for the heterophasic PP copolymer should be given. Table 1 is confusing. What is the definition for the “Lab scale compounds derived from MBs”. How is the formulations developed under this subject? For example, what is the difference between FR1 and FRMB1? It is reported in the text” Accordingly (Table 1), six new FR compounds (FRMB3-FRMB8) were developed based on MB1 and two (FRMB9, FRMB10) based on MB2, via twin-screw extrusion of the desired amounts of reference PP and the required MB” however Table 1 itself is confusing. Why is there such a high amount of additives in industrial scale MBs? Description for ADD1, ADD2, ADD3, ADD4 and Bromine content analysis should be given under the Table as a footnotes. The chemical structures of these compounds can also be given in supporting info.

Response 2:

  • Composition of the heterophasic PP is not given by the manufacturer (Repsol) and could not be detected and quantified in the DSC analysis (cooling and 2nd heating curves) in Figure 2a,b. Typically, the iPP block-copolymers comprise ethylene-propylene monomer (EPM) or ethylene-propylene-diene monomer (EPDM). This was Added in the text and highlighted in yellow (Lines 142-143).
  • Table 1 is the outline of all formulations developed within this article. It was divided due to reviewers’ confusion in Table 1 and Table 2. In Table 1, “Lab-scale compounds” are FR1, FR2: reference PP (ISPLEN, PB131N5E) was directly melt-mixed with the respective amount of additives (according to the formulations given in Table 1) in the lab-scale twin-screw extruder as described in chapter 2.2 Compounding. In Table 1 also the “Industrial Scale Masterbatches” are MB1, MB2: they were prepared in the industrial scale twin-screw extruder by PLASTIKA KRITIS SA (ca. 100 kg each) and further supplied to us. In these masterbatches PP carrier (ECOLEN HZ40S) was the polymer in which a much higher additive content (according to each formulation) was melt-mixed. This was added to the text and was highlighted in yellow to guide the reader better. In Table 2, in the case of the “Lab-scale compounds derived from MBs” PP (ISPLEN, PB131N5E) is not directly melt- mixed with the pure additives as in the case of FR1, FR2; it is melt-mixed with the desired MB amount, in order to mimic the conduit manufacture process. This means that FRMB1, FRMB2 exhibit exactly the same additive concentrations as FR1 and FR2, but they were developed by mixing the polymer with the respective amount of MB. This masterbatch based compounding ensures a better and more homogenous dispersion of the additives as proved in Figure 10 for the impact strength. Changes were made in the text in lines 169-176.
  • The adoption of the Masterbatch technology offers high flexibility in the end-users (in this case conduit manufacturers), since they can re-adjust very easily the concentration of the contained additives in their production line. Conduit manufactures (in our case KOUVIDIS SA) do not perform compounding in their production lines, i.e., direct melt-mixing of the desired additives with the mother polymer and they do not have the expertise or the proper extrusion equipment to do so. What they typically do is to order to a compounder (in our case PLASTIKA KRITIS SA), i.e., a manufacturer of masterbatches, a masterbatch that includes the desired functionalities, e.g. flame retardants, UV stabilizers, pigments etc. Conduit manufacturers require very high concentrations in the masterbatch, so as to have during the dilution process a ratio of 1:9 MB:PP or even higher and the final product is not affected by the presence of the polymer carrier in the MB. In any case, the concentration of the industrial masterbatches MB1 and MB2 were specially designed according to the expertise of our industrial compounder (PLASTIKA KRITIS SA) and the requirements of the conduit manufacturer (KOUVIDIS SA). They were based on the initial FR1, FR2 formulations (4 times and 10 times the concentration of FR1 and FR2 respectively), so as to have a total additive concentration in the range of ca. 40%, while maintaining the ratio between the individual additives, i.e. ADD1:ADD2 = 10:1 in MB1 and ADD3:ADD4 = 7:1.
  • Description for ADD1, ADD2, ADD3, ADD4 and Bromine content analysis were given under Table 1 as footnotes. The chemical structures of all additives are given in Figure S1 in supporting info section.

Comment 3: Figure 3. inset data is not read well.

Response 3: Figure 3 was updated in higher resolution in the text so as to be clearly read.

Comment 4: An evaluation was made about additives compatibility with the polymer matrix between line and their effects on mechanical properties were examined. Similarly, was there a deviation in the test samples in the FR tests?

Response 4: The deviation of burning times for FR1 and FR2 were not that significant. Especially in the case of FR1 the specimens were very reproducible (see table below). In the case of FR2 there was some deviation especially during the 2nd afterflame time but this is also connected to the strong flame dripping. The optimum dispersion of the additives is not so critical for the FR performance as it is for the mechanical properties and especially for the impact strength test. As mentioned in the text, the fact that ADD1 and ADD2 could melt during compounding and/or processing of the specimens, would create local weak spots within the specimen, thus affecting the mechanical behavior. This is not the case upon burning of the sample, since in these weak spots the FR additives would still react properly and will terminate fire.

UL94 specimens

1st afterflame time (s)

2nd Afterflame time (s)

Total burning time (s)

Dripping (Y/N)

Cotton Ignition (Y/N)

UL94 Class

FR1

1

1.26

1.46

2.72

Y

N

V0

2

1.13

1.82

2.95

Y

N

V0

3

2.22

1.11

3.33

Y

N

V0

4

1.04

1.14

2.18

Y

N

V0

5

1.22

1.3

2.52

Y

N

V0

FR2

1

1.94

3.02

4.96

Y

Y

V2

2

1.87

4.42

6.29

Y

Y

V2

3

1.83

2.51

4.34

Y

Y

V2

4

1.8

3.12

4.92

Y

Y

V2

5

1.74

3.15

4.89

Y

Y

V2

Comment 5: Figure 8 and Figure 9: inset data is not read well.

Response 5: Figures 8 and 9 were updated in higher resolution in the text so as to be clearly read.

Reviewer 2 Report

Comments and Suggestions for Authors

The article touches on the interesting topic of fire safety of polymer materials. However, the way the text is written makes the article difficult to read. I suggest that the authors sort out some stylistic issues and try to make it easier for the reader to understand. Due to this and the following comments/questions, I cannot recommend the article for publication and I suggest its major revision.

Questions/suggestions:

1. In some places in the article there are errors in the cited literature (lack of attribution of literature items). Please correct it.

2. Line 379 - 381 and Fig. 2a - I do not agree with the statement that the curve of the FR2 sample is close to the PP curve. The changes obtained are at the same level as in the case of the FR1 sample, e.g. Tc shift.

Fig. 2b. - In the case of sample FR2, there is an obvious additional arm on the melting peak. What could be the reason for this change?

Please improve the description and analysis of the DSC test of the FR2 sample.

3. Figure 3 is unreadable. Please fix it.

4. Line 446 – 471. The presented description of the burning study does not correspond to the results presented in Figure 4. This is especially visible in the case of the FR2 sample, where changes in the recorded parameters look completely different than those described by the authors.

5. Line 467 - 469. Authors state: “All aged specimens exhibited a 467 small increase in the smoke emission during the UL94 test….”. Please explain the reason for the increase in smoke emissions as a result of the samples' age.

6. What could be the reason for the observed increase in tensile strength? In most cases, the introduction of a powder filler into PP causes a decrease in the tensile strength of this polymer.

7. Did pure polypropylene break during the impact test? Specimens without notches were used, suggesting that the specimens did not break. Therefore, providing the impact strength value in the case of PP should be accompanied by an appropriate comment.

8. Again, as in the case of tensile strength, I am curious about the lack of reduction in impact strength after the introduction of flame retardant compounds (which is a typical phenomenon)? What could be the reason for the lack of impact strength reduction?

9. Figure 8 is unreadable. Please fix it.

10. Line 614-618 - On what basis was the content of FR1 and FR2 in the masterbatch determined?

11. Fig 10. Once again - reason of the lack of decrease in impact strength after the introduction of the filler. Did the authors consider that differences in impact strength may result from adding a different type of PP?

12. Why was the FRMB7 sample selected as a material for industrial scale? Do the presented results contradict this choice?

13. Table 2 - On what basis was the MB content determined?

Author Response

The article touches on the interesting topic of fire safety of polymer materials. However, the way the text is written makes the article difficult to read. I suggest that the authors sort out some stylistic issues and try to make it easier for the reader to understand. Due to this and the following comments/questions, I cannot recommend the article for publication and I suggest its major revision.

Questions/suggestions:

Comment 1: In some places in the article there are errors in the cited literature (lack of attribution of literature items). Please correct it.

Response 1: Reference list was updated (rearrangement of reference order from [38] to [40]) and proper replacement within the text has also been performed. All changes in the reference order were highlighted in yellow in the text.

Comment 2: Line 379 - 381 and Fig. 2a - I do not agree with the statement that the curve of the FR2 sample is close to the PP curve. The changes obtained are at the same level as in the case of the FR1 sample, e.g. Tc shift. Fig. 2b. - In the case of sample FR2, there is an obvious additional arm on the melting peak. What could be the reason for this change? Please improve the description and analysis of the DSC test of the FR2 sample.

Response 2:

  • The reviewers’ comment regarding Tc shift is not very clear to us. Apart from to Fig. 2a, already in the manuscript there is Table S1 (Supplementary Info section), where DSC numerical data are provided: Tc in FR1 shifts ca. 7 °C lower compared to PP, which is a significant difference, while in the case of FR2, the Tc shift is ca. 0.5 °C higher compared to PP, which is not that significant and also it is within the error of the DSC instrument. It seems that the effect of the additives contained in FR1 on Tc and in all thermal properties (Tm2, Xc) is much more pronounced than in FR2. Text was slightly modified and was highlighted in yellow.
  • This is an important remark, and could probably show the melting of smaller crystals. Moreover, this is arm is not connected to the contained FRs (ADD3 and ADD4), since they are considered infusible according to their manufacturer (Italmatch Chemicals) and according to their DSC curve (Figure S2), where no melting endotherm is observed at the specific temperature range. A small comment was added in the text to emphasize this remark in lines 374-377.

Comment 3: Figure 3 is unreadable. Please fix it.

Response 3: Figure 3 was updated in higher resolution in the text so as to be clearly read.

Comment 4. Line 446 – 471. The presented description of the burning study does not correspond to the results presented in Figure 4. This is especially visible in the case of the FR2 sample, where changes in the recorded parameters look completely different than those described by the authors.

Response 4: For our understanding the comment of the reviewers concerns the ups and downs observed in the total burning time observed for FR2 specimens during ageing. However, it should be noted that in the specific test, V2 classification was maintained and this highlighted in the text. From our experience in the field, when intense flame dripping is part of the FR mechanism, as it is in FR2 such variations in the total burning time and further in the mass loss of the burnt specimen is a typical phenomenon. In any case this was already mentioned in the text FR2 also retained successfully the V2 category, but exhibited an increasing trend in total burning time”. To further improve it the text was updated and was highlighted in yellow so as to guide the reader better (Lines 451-469).

Comment 5: Line 467 - 469. Authors state: “All aged specimens exhibited a small increase in the smoke emission during the UL94 test….”. Please explain the reason for the increase in smoke emissions as a result of the samples' age.

Response 5: This was a qualitative macroscopical observation and could be attributed to minor ageing that the FR additives suffered, macroscopically observed in the case of FR1, via the occurrence of stickiness. However, the retention of the UL94 classification in both compounds, proves that this ageing of the additives was not that significant. A small comment was added in the text and was highlighted in yellow (lines 470-471). The actual smoke density of these FR systems in the form of corrugated conduits was quantified accurately according to EN IEC 61034-2 standard (Table 3).

Comment 6: What could be the reason for the observed increase in tensile strength? In most cases, the introduction of a powder filler into PP causes a decrease in the tensile strength of this polymer.

Response 6: According to Table S2, the tensile strength (σmax) of PP was determined at 25.1 MPa prior to any ageing, while in the case of FR1 it was found slightly lower (ca. 3%), i.e. 24.4 MPa and in the case of FR2 higher, i.e. 28.2% (ca. 12%). The incorporation level of the additives is rather low (11 and 4% respectively for FR1 and FR2), for exhibiting significant changes in terms of σmax. The impact of additives incorporation is much more pronounced prior to ageing in terms of elongation (εmax) and Young’s Modulus (E), where a strong decrease (ca. 40%) in εmax and an increase in E (10-20%), proving that the FR compounds became stiffer and more brittle than the reference PP. Significant reduction in terms of impact strength are indeed observed when FR loading is in the range of 20-30%, mostly when intumescent system FRs are used. Actually, we have tested such FR systems, but their mechanical properties loss rendered them inappropriate for the application of conduits, therefore were not included in the current article.

Comment 7: Did pure polypropylene break during the impact test? Specimens without notches were used, suggesting that the specimens did not break. Therefore, providing the impact strength value in the case of PP should be accompanied by an appropriate comment.

Response 7: All impact specimens were measured unnotched. Accordingly, for reference PP all measured specimens, prior to and after weathering, did not break. Similar behavior was observed for FR2, prior to and after weathering, where loading level of additives is very low (4%). On the contrary, FR1 presented a totally different behavior, with almost all specimens fully broken prior to and after weathering, highlighting even more the occurrence of weak spots as a consequence of potential melting of the additives during processing and/or specimen preparation, as commented also in the text. This was updated in the text and was highlighted in yellow (lines 503,509).

Comment 8: Again, as in the case of tensile strength, I am curious about the lack of reduction in impact strength after the introduction of flame-retardant compounds (which is a typical phenomenon)? What could be the reason for the lack of impact strength reduction?

Response 8: As mentioned in Q6, the incorporation level of the pertinent FR additives is rather low (11 and 4%) to cause significant changes in impact strength, therefore both compounds exhibit an average value close to PP (prior to ageing). Indeed, in other FR systems comprising intumescent additives that we have also checked within this project (not mentioned in this work), used at a loading level of 20-30% the impact strength reduction was severe, i.e. decrease reaches even 45%.  

Comment 9: Figure 8 is unreadable. Please fix it.

Response 9: Figures 8 and 9 were updated in higher resolution in the text so as to be clearly read.

Comment 10: Line 614-618 - On what basis was the content of FR1 and FR2 in the masterbatch determined?

Response 10: First of all, FR1 and FR2 compounds were developed in the lab-scale, via direct melt mixing of the additives with the reference PP. All the lab-scale experimental work presented in the article, prior to and after ageing tests was performed in order to assess the potentiality of these compounds for the application of conduits. Our research project involved the adoption of these promising compounds in the industrial scale and the real production of corrugated conduits from our end-user (KOUVIDIS SA). Conduit manufacturers always work with masterbatches (MBs), so that they can easily adjust the desired additive loading in their production lines by applying different MB:PP ratios. According to our end-user (KOUVIDIS SA), the desired MB:PP ratio in the conduit production line is typically in the range of 1:9 or even higher. Moreover, according to our industrial compounder (PLASTIKA KRITIS SA) MBs are typically prepared in high additive concentration (e.g. ca. 40%). In any case, the concentration of the industrial masterbatches MB1 and MB2 were specially designed according to the expertise of our industrial compounder (PLASTIKA KRITIS SA) and the requirements of the conduit manufacturer (KOUVIDIS SA). They were based on the initial FR1, FR2 formulations (4 times and 10 times the concentration of FR1 and FR2 respectively), so as to have a total additive concentration in the range of ca. 40%, while maintaining the ratio between the individual additives, i.e. ADD1:ADD2 = 10:1 in MB1 and ADD3:ADD4 = 7:1. A small sentence was added in the text and was highlighted in yellow in order to clarify this (lines 617-621).

Comment 11: Fig 10. Once again - reason of the lack of decrease in impact strength after the introduction of the filler. Did the authors consider that differences in impact strength may result from adding a different type of PP?

Response 11: As already explained in Q6 and Q8, the additive loading is very low to cause significant decrease in the mechanical behavior. Figure 10 shows that melt mixing of the Industrial Scale MBs with PP resulted in a more homogenous dispersion of the additives in FMRB1, FRMB2, in comparison to the original lab scale-compounds FR1, FR2, where the pure additives were directly melt-mixed with PP. The average impact strength values were found slightly increased for FRMB1 and FRMB2, but what was really improved was the RSD value, which was limited to 2.1% in FRMB1 (from 21.3 in FR1) and 4.1% for FRMB2 (from 13.8 in FR2), clearly proving that the specimens are much more homogenous. Moreover, the addition of a different PP type, i.e. PP carrier, in FRMB1 and FRMB2 could not cause the increased impact strength, since according to our measurements (not presented in the article), PP carrier shows an impact strength of ca. 24.5 kJ/m2, much lower than the value of reference PP (59.3 kJ/m2).  

Comment 12: Why was the FRMB7 sample selected as a material for industrial scale? Do the presented results contradict this choice?

Response 12: FRMB7 was selected from our end-user (KOUVIDIS SA), mostly for cost-effective reasons, as mentioned also in the text (This attempt aims mainly at a cost-effective approach for industrializing these compounds). ADD1 and ADD2 are really expensive additives, thus resulting in a very high price of MB1 (ca. 36€/kg), while other commercially available FR MBs (mostly halogenated) according to our end-user cost ca. 10-12 €/kg. In fact, the lab-scale tests for producing the FRMB3-FRMB10 compounds aimed exactly at minimizing the FR content until the compound still reaches a classification in UL94, thus avoiding additional testing in the industrial scale. FRMB7 of course resulted in V2, which is worse than the initial V0 of FR1 and FRMB1, yet the overall performance of conduit C1 (which was manufactured according to FRMB7), in terms of flame propagation, resistance to impact and smoke density complies with EN IEC 61386-22:2021 and EN IEC 61034-2.

Comment 13: Table 2 - On what basis was the MB content determined?

Response 13: Please see our answer in Q10 and Q12. Table 2 was updated to Table 3. Therein the value of 8% of MB1 or MB2, means that the conduit was manufactured by melt mixing 8% of MB1 or MB2 with 92% of PP. The actual additive content of the MBs is given already from Table 1, i.e., 44% for MB1 and 40% for MB2. Some grammatical changes were made in Table 3 in order to guide better the reader.

Round 2

Reviewer 2 Report

Comments and Suggestions for Authors

The authors answered the submitted questions in detail. The answers are comprehensive and dispel my doubts. I am in favor of publishing the article. However, some minor editorial changes are still needed:

1. Lines 298-299, 542-543, 622-623 - there is still a citation error.

2. Figures 3, 7, 8 and 9 - require changes. They are too small and difficult to read even after attaching higher resolution drawings.

Author Response

The authors answered the submitted questions in detail. The answers are comprehensive and dispel my doubts. I am in favor of publishing the article. However, some minor editorial changes are still needed:

We updated all minor editorial changes suggested by the reviewer and also improved some linguistic terms. All changes made for this revision were highlighted in light green in the revised manuscript text.

  1. Lines 298-299, 542-543, 622-623 - there is still a citation error.

We went through the text again and again in detail and we corrected all citation errors.

  1. Figures 3, 7, 8 and 9 - require changes. They are too small and difficult to read even after attaching higher resolution drawings.

Figures 3, 7, 8 and 9 were again updated, with higher resolution and higher font size so as to be more clearly read.
